# Automated Parallel Dialysis for Purification of Polymers

**DOI:** 10.3390/polym14224835

**Published:** 2022-11-10

**Authors:** İpek Terzioğlu, Carolina Ventura-Hunter, Jens Ulbrich, Enrique Saldívar-Guerra, Ulrich S. Schubert, Carlos Guerrero-Sánchez

**Affiliations:** 1Laboratory of Organic and Macromolecular Chemistry (IOMC), Friedrich Schiller University Jena, Humboldtstrasse 10, 07743 Jena, Germany; 2Polymerization Processes Department, Centro de Investigación en Química Aplicada (CIQA), Blvd. Enrique Reyna No. 140, Saltillo 25294, Coahuila, Mexico

**Keywords:** polymer libraries, purification, automation, dialysis, high-throughput/-output experimentation

## Abstract

The implementation of a dialysis method for the simultaneous purification of different polymer materials in a commercially available automated parallel synthesizer (APS) is discussed. The efficiency of this “unattended” automated parallel dialysis (APD) method was investigated by means of proton nuclear magnetic resonance (^1^H-NMR) measurements, which confirmed that the method enables the removal of up to 99% of the unreacted monomer derived from the synthesis of the corresponding polymers in the APS. Size-exclusion chromatography (SEC) revealed that the molar mass and molar mass distribution of the investigated polymers did not undergo significant changes after the application of the APD method. The method discussed herein can be regarded as a good alternative to the “unattended” and reliable purification of polymer libraries prepared in APS. This method may be useful for overcoming current limitations of high-throughput/-output (HT/O) synthesis of polymer libraries, where purification of the generated materials currently represents a significant constraint for establishing fully automated experimental workflows necessary to advance towards a full digitalization of research and development of new polymers for diverse applications.

## 1. Introduction

Current digitalization tendencies in the research and development (R&D) of polymer materials have gained importance in recent years [1]. Access to fully digitalized experimental protocols would enable significantly increased efficiency in the discovery and application processes of (new) polymers. To this end, a successful digitalization of the polymer R&D process will undoubtedly require automating and/or parallelizing in a reliable manner different experimental tasks, such as synthesis, characterization, purification, and testing of polymer materials. First approaches toward this ultimate goal have shown progress in establishing efficiently automated and/or parallel experimental protocols for the synthesis and characterization of polymer libraries employing combinatorial, high-throughput/-output (HT/O), or flow chemistry techniques [2,3,4]. Nonetheless, HT/O experimentation, as used nowadays in different polymer laboratories, currently faces obstacles that need to be addressed to contribute to the establishment of fully automated and/or parallelized experimental workflows necessary to progress the digitalization of R&D in this field. An important bottleneck in this regard is the lack of automated and/or parallel techniques for purifying synthesized polymer materials and/or libraries for further and reliable testing. Thus, the development of suitable and efficient automated and/or parallel purification procedures for polymer libraries, as well as their integration into fully automated experimental workflows, is still a task mostly unsolved and overlooked in many polymer laboratories, which hinders the full potential of HT/O experimentation and its efficient utilization in digitalizing polymer R&D [3]. Access to pure polymers (i.e., free of residual monomers, organic solvents, and/or other additives necessary for their manufacture) is, in many cases, required to carry out fundamental research (e.g., characterization or post-modifications) and/or application-related tests. For instance, it has been reported that traces of impurities could affect the physical properties of substances, such as melting, freezing, and boiling points [5], as well as thermal transitions [6]. Additionally, such contaminations can also show undesired plasticizing effects in materials [7] or undesired (cyto)toxic effects during the evaluation of polymers for biomedical applications [8]. Furthermore, residual monomers can also negatively impact polymer properties in terms of biocompatibility [9,10], mechanical properties [11], the surface porosity of materials [12], degradation [13], and in other applications, such as drug delivery, where they can cause the so-called initial burst effect [14] or show undesired interactions with drugs [15]. Consequently, to prevent these effects, polymers might be purified by diverse methods such as dialysis [16,17], precipitation [18], chromatography [19], use of supercritical fluids [20], gas or steam stripping [21], and physisorption [8]. A recent report summarizes and compares the advantages/disadvantages of some of the polymer purification methods most commonly utilized in R&D [8]. Despite the different alternatives available for purifying polymers, dialysis still stands as a widely used technique to selectively separate residual monomers and other chemical substances (i.e., residual solvents and/or oligomers) from polymer solutions through a semipermeable membrane. Moreover, dialysis is not only applicable for purifying synthetic macromolecules, but it is also commonly used to purify biopolymers, such as sugars [22,23] and proteins [24,25] or drugs [26,27].

However, the rapid and effective purification of polymer libraries obtained from HT/O synthesis still remains an unsolved experimental task, which is labor-intensive and time-consuming, without mentioning the very often overlooked lack of repeatability and reproducibility of the obtained purified polymers [8]. This aspect actually represents a significant constraint for polymer-related R&D workflows. For instance, recent contributions using HT/O experimentation to screen generated polymer libraries still suffered from this unsolved drawback as the purification of the materials is still manually performed using a one-at-a-time approach [28] or overlooked [29]. In certain R&D projects, this drawback could lead to inconclusive ambiguous results due to the evaluations of poorly reproducible or impure polymer candidates.

To overcome this obstacle, recent research efforts have focused on automating the dialysis process using different approaches [30]. For instance, dialysis for a limited number of proteins has been reported [31]. Schuett et al. have recently developed a one-at-a-time semi-automated polymer purification method based on dialysis with an inline proton nuclear magnetic resonance (^1^H-NMR) capability to monitor the removal of impurities in real time [32,33]. In a follow-up investigation, Schuett et al., 2022 reported an automated parallel multi-step synthesis of block copolymer libraries that employed their previously developed purification method [34]. All in all, the number of chemical samples that can be dialyzed either in an automated or manual fashion in a certain period of time still remains limited to keep up with the current needs of HT/O synthesis of polymer libraries utilized in R&D.

Hence, this contribution describes the implementation of a dialysis method in a commercially available automated parallel synthesizer (APS) to perform the unattended and simultaneous purification of multiple polymer samples or libraries. The automated parallel dialysis (APD) method proposed herein can be regarded as an alternative to overcome the current limitations in HT/O synthesis of polymer libraries, where purification of the generated materials significantly limits the establishment of fully automated experimental workflows necessary to advance towards the full digitalization of R&D for new polymers with diverse applications.

## 2. Materials and Methods

### 2.1. Materials

4-Cyano-4-(thiobenzoylthio) pentanoic acid (CPAD, 97%), 4,4′-azobis(4-cyanopentanoic acid) (ACVA, ≥98%), poly(ethylene glycol) methyl ether methacrylate (PEGMA, *M*_n_ 500 g·mol^−1^), *N*,*N*-dimethyl acrylamide (DMA, 99%), 2,2′-azobis(2-methylpropionitril) (AIBN, 98%), and 1,3,5-trioxane (≥99%) were obtained from Aldrich (Saint Louis, MO, USA) and used as received. 2-(Dodecylthiocarbonothioylthio)-2-methylpropionic acid (DTMPA, 97%) was purchased from Strem Chemicals (Boston, MA, USA) and utilized as received. Glycerol mono-methacrylate (GMMA, 95%) was acquired from Polysciences, Inc. (Warrington, PA, USA), and acrylic acid (AA, 99.5%) was obtained from Alfa Aesar (Tewksbury, MA, USA), and used as received. All monomers were purified by stirring in the presence of inhibitor-remover beads (for hydroquinone and monomethyl ether hydroquinone, Aldrich, Saint Louis, MO, USA). 1,4-Dioxane (99.5%) was purchased from Acros Organics (Geel, Belgium), and used as received. Ethanol was provided by the solvent purification system MB-SPS-800 (MBraun, Stratham, NH, USA) and stored under argon. Float-A-Lyzer^®®^ G2 dialysis tubes (0.5 to 1 kDa, d_membrane_ = 10 mm, volume = 10 mL) were acquired from Repligen (former Spectrum Labs, Waltham, MA, USA) and used for the dialysis of the polymer solutions.

### 2.2. Reversible Addition–Fragmentation Chain-Transfer (RAFT) Polymerizations

RAFT homopolymerizations of the PEGMA, GMMA, DMA, and AA monomers were performed in microwave vials. Table 1 summarizes the reagents, amounts, and reaction conditions utilized for the synthesis of the investigated homopolymers. Note that the solubility and reactivity of a RAFT agent mainly depend on the R and Z groups; as such, different RAFT agents are more suitable for specific types of monomers, as addressed elsewhere [35,36,37]. In general, CPAD is a suitable RAFT agent to polymerize methacrylate- and methacrylamide-based monomers, but it is less effective for the polymerization of acrylic- and acrylamide-based monomers; for the latter, a better selection would be DTMPA [37]. Additionally, 1,3 trioxane, an inert compound, was added to the polymerization reactions (at a concentration of ca. 10 mg mL^−1^ referred to the total reaction volume) to be used as an internal reference [38,39,40,41] to monitor monomer conversion via ^1^H-NMR analyses. After preparation, the reaction mixtures were degassed for 30 min by sparging nitrogen gas and subsequently placed into a preheated oil bath at the desired temperature. After the predetermined reaction time elapsed, the reaction mixtures were cooled to room temperature and exposed to air. Thereafter, the crude polymerization solutions were placed in the proposed APD system to monitor the removal of residual monomer in time (see details in the next section). The dialyzed polymers were lyophilized for 48 h. The obtained dry polymers were identified as follows: Poly(poly(ethylene glycol) methyl ether methacrylate) (P1; P(PEGMA)), poly(glycerol mono-methacrylate) (P2; P(GMMA)), poly(*N*,*N*-dimethyl acrylamide) (P3; P(DMA)), and poly(acrylic acid) (P4; P(AA)). Schematic representations of the chemical structure of synthesized polymers are displayed in Figure 1. ^1^H-NMR spectra of the obtained, dialyzed and dried polymers are displayed in Appendix A. Table 2 also summarizes monomer conversion (estimated from ^1^H-NMR measurements by integrating the signals of the vinylic protons of the respective monomers) and molar mass of the synthesized polymers (estimated from size-exclusion chromatography (SEC) analyses) before and after purification.

### 2.3. Characterization 

#### 2.3.1. Size-Exclusion Chromatography (SEC)

The SEC measurements of P1 to P3 were performed on an Agilent system equipped with a G1329A Autosampler and a G1310A pump. A G7162A refractive index (RI) detector and PSS GRAM (Polymer Standards Service GmbH, Mainz, Germany) column with dimethylacetamide (DMAc) + 0.21 wt. % LiCl as eluent with a flow rate of 1 mL min^−1^ at 40 °C. The relative number average molar mass (*M*_n_) of the polymers was estimated using a calibration curve prepared from poly(methyl methacrylate) (PMMA) standards of low dispersity (*Đ*). The SEC measurements of P4 were analyzed in a Jasco system equipped with a PU-980 pump. A RI-2031 Plus RI detector and PSS SUPREMA (Polymer Standards Service GmbH, Mainz, Germany) column with an aqueous solution of 0.1 M NaNO_3_ + 0.05% NaN_3_ as mobile phase with a flow rate of 1 mL min^−1^ at 30 °C. The relative molar mass of P4 was estimated using a calibration curve prepared from poly(ethylene glycol) (PEG) standards of low dispersity.

#### 2.3.2. Proton Nuclear Magnetic Resonance (^1^H-NMR) Spectroscopy

Proton Nuclear Magnetic Resonance (^1^H-NMR) spectroscopy was selected to monitor the monomer conversion during the RAFT polymerizations and residual monomer during the APD procedure. The spectra were recorded at room temperature using a 300 MHz Bruker Avance I spectrometer equipped with a dual ^1^H and ^13^C probe head and a 120 × BACS automatic sample changer. The monomer conversion of P1 and P3 was measured in deuterated chloroform (CDCl_3_), whereas for P2 and P4 in dimethyl sulfoxide (DMSO-*d_6_*). A ^1^H-NMR method with a water-suppression function was utilized to monitor and quantify the residual monomers during the APD process using deuterium oxide (D_2_O).

### 2.4. VMR PP 4083 Peristaltic Pump

A VMR PP 4083 series peristaltic pump (Avantor Sciences, Radnor, PA, USA) was used for filling (and/or emptying) the dialysis container. Multichannel (up to four channels) and simultaneous feeding are provided by this pump. To avoid the contamination of the dialysis medium and the entire pumping system, the utilized tubing was only in contact with deionized water. The utilized 4 L dialysis container was pre-filled in 25 min using this peristaltic pump at a flow rate of 160 mL min^−1^.

### 2.5. Chemspeed ASW2000 Automated Parallel Synthesizer (APS)

A commercially available automated parallel synthesizer (APS) (Chemspeed ASW 2000, Fuellinsdorf, Switzerland) was used to implement the proposed APD method. The ASW2000 APS is equipped with a robotic arm bearing a needle head (NH) connected to tubing and to two dilutor pumps for handling liquids automatically, interchangeable reactor blocks, 96-well plate racks, sample vial racks, and a stock solution rack. The reactor blocks can consist of sixteen or four glass reaction vessels (13 mL or 75 mL, respectively) with thermal jackets connected in series to a heating/cooling system (from −20 to 140 °C, Huber Ministat CC 3, Baden-Wurttemberg, Germany), and a vortex mixer (up to 1400 rpm). Additionally, each reaction vessel is equipped with a cold-finger reflux condenser (5 °C). The automated liquid handling system is used for dispensing reagents and sampling into/from the reactor vessels and/or sample vials. The NH is connected to a solvent reservoir bottle for needle rinsing after each liquid transfer. Further details about the characteristics and capabilities of the Chemspeed ASW 2000 platform can be found elsewhere [42,43,44]. For the proposed APD method (Figure 2), a custom-made rack (Figure 2D) was employed to house the utilized and commercially available Float-A-Lyzer^®®^ G2 dialysis tubes (Figure 2C). The rack containing the dialysis tubes was placed between a reactor block and the stock solution rack on the deck of the APS, as depicted in Figure 2A,B. Utilizing the software of APS, multiple parallel synthesis and/or parallel dialysis can be programmed and executed in an unattended fashion. Sample dilutions and/or transfers of polymerization mixtures, for instance, from the reaction vessels into the dialysis tubes, can be automatically performed with the aid of the liquid handling system of the robotic platform. It is also possible to automatically and periodically monitor the progress of the performed chemical reaction and/or dialysis purifications at different time intervals by withdrawing aliquots from the reaction vessels and/or dialysis tubes followed by suitable analyses (e.g., ^1^H-NMR or SEC measurements); the sample preparation for such an analysis can also be performed in an unattended fashion [42,43,44].

## 3. Results and Discussion

The setup for the proposed APD method was implemented into a commercially available APS (Chemspeed ASW2000), as depicted in Figure 2A,B. First, two dialysis racks were designed and manufactured to house a predetermined number of Float-A-Lyzer^®®^ G2 dialysis tubes (0.5 to 1 kDa, d_membrane_ = 10 mm, volume = 10 mL; Figure 2C). The dialysis tubes were placed and fixed between the two racks to allow the liquid handling system of the APS to periodically withdraw aliquots from the tubes during the dialysis process for subsequent analyses. Note that the utilization of two racks (one on top of the other one) was necessary to prevent the NH of the APS from lifting the tubes during the sampling (Figure 2D). Furthermore, it was initially observed that the initial volume of the solutions to be purified increased during dialysis due to intrinsic osmotic pressure. Consequently, the initial reaction volume in each dialysis tube was set at 6 mL to prevent polymer solutions from overflowing out of the open dialysis tubes. According to the manufacturer, the utilized tubes are appropriate for dialysis using solvents such as methanol, ethanol, water, and/or mixtures derived thereof. Hence, an intrinsic limitation of the APD method discussed herein is that polymers to be subjected to this proposed purification method must be soluble in the aforementioned solvents. In the proposed setup, the container housing the selected solvent to perform dialysis can be filled up in different ways using a suitable pump (or manually), which includes recirculation mode, full exchanges of solvent at predetermined times, or a single fill. Hence, potential users of the APD method can choose and optimize the dialysis conditions according to their needs. In this contribution, the dialysis container was filled up with the aid of a peristaltic pump (VWR PP 4083) at 200 RPM (or manually) with 4 L of water. Additionally, the dialysis solvent was continuously mixed by means of a magnetic system (700 RPM) placed at the bottom of the container.

In our experiments, the solvents (dioxane or ethanol) used for the RAFT polymerization reactions were partially removed by applying a stream of compressed air into the vessels containing the respective raw polymerization reactions. Thereafter, the obtained mixtures containing the polymer, residual monomer, and the remaining solvent were diluted with deionized water to obtain 6 mL of solution. The diluted polymer solutions were automatically (or manually) transferred into the respective dialysis tubes housed at the customized rack in the APS. Next, the dialysis container was filled up with 4 L of deionized water. The dialysis tubes remained open throughout the entire purification procedure to facilitate the sampling and to release the built osmotic pressure. Aliquots were withdrawn from the dialysis tubes at predetermined times with the aid of the automated liquid handling system of the APS for SEC and NMR analyses. The software program of the APS utilized for the described APD procedure can be found in Appendix A. Once the purification process was completed, dialyzed polymer solutions were transferred from the dialysis tubes into suitable vials with the aid of the automated liquid handling system of the APS.

Thus, biocompatible and water-soluble polymers (P1–P4; Table 1 and Table 2) synthesized by the RAFT technique were purified by the proposed APD method. ^1^H NMR spectra of the dialyzed and dried polymers are available in Appendix A. To evaluate the effectiveness of the proposed APD method, the amount of residual monomer removed during the APD process was estimated by means of ^1^H-NMR analyses. For this purpose, the integral value of the signal corresponding to the vinylic protons of the respective monomer has been compared: (1) to the integral value of the signal corresponding to the aromatic protons of the RAFT end group at 7.38 to 7.98 ppm (5H) for polymers P1 and P2 or (2) to the integral value of a signal corresponding to the backbone of the synthesized polymer at 1.38–1.88 ppm and 2.11–2.54 ppm for P3 and P4, respectively. For instance, Figure 3A displays the ^1^H-NMR spectra of the aliquots withdrawn at different times during the APD process applied to P4. Furthermore, Figure 3B shows the decrease in residual monomer content vs. dialysis time for all four polymers subjected to the APD method. Note that for the removal of the PEGMA and GMMA monomers, longer dialysis times were required to reach similar residual monomer contents to those obtained for the AA and DMA cases in shorter periods of time. This observation could be correlated to the size/molar mass of the respective monomer molecules (Figure 1) where the diffusion process of larger molecules through the dialysis membrane might be slower as compared to that of molecules of smaller size or molar mass (i.e., PEGMA > GMMA > DMA > AA (see Figure 1)). Based on this assumption, one can already anticipate that monomers such as PEGMA or GMMA will require longer dialysis times at certain given conditions. In general, the physicochemical properties of the monomers and respective polymers will dictate the selection of the type of dialysis membrane as well as the dialysis conditions (i.e., time, solvent, etc.).

As shown in Table 2, the *M*_n_ and *Đ* values of polymers P1-P4 were determined by SEC analyses before and after the application of the APD method. For polymers P1, P3, and P4, no significant changes were observed in the *M*_n_ and *Đ* values (Table 2) and for the corresponding molar mass distributions (SEC traces displayed in Figure 4). In contrast, a noticeable increase in the *M*_n_ and *Đ* values (Table 2) and modification of the molar mass distribution towards higher molar mass values (Figure 4) were observed for polymer P2. This observation may be ascribed to the rise of possible interactions between P(GMMA) polymer chains via hydrogen bonding during the dialysis process, as suggested elsewhere [43], which may promote the formation of polymer species of different molar mass and, therefore, the modification of the hydrodynamic volume of such polymer chains during the respective SEC analyses. This behavior can also be related to the presence of water traces (in our case, most likely derived from the dialysis process), which may remain in the respective polymer samples, as suggested elsewhere [45,46].

It is worth mentioning that the monomer conversion of the investigated RAFT polymerizations reported in Table 2 was deliberately kept at a relatively “low” level (i.e., <60%) because of the two main following reasons: (1) To prove that the proposed APD method is still suitable to remove relatively high concentrations of residual monomers and impurities, and (2) to maintain high levels of RAFT end-group fidelity of the synthesized homopolymers in order to establish, in the near future, a fully automated workflow for the synthesis of block copolymer (and/or other functionalized polymer) libraries. Nonetheless, the proposed APD method should be fully applicable (and, in fact, more easily applicable) to RAFT polymerization systems where the achieved monomer conversion levels are >80% (or even >90%). In fact, a faster monomer removal for such “high” monomer conversion levels could be expected (as compared to RAFT systems where monomer conversion is <60% (or even <50%)) since the initial concentration of residual monomer (i.e., before dialysis and after the polymerization reaction) would be lower.

## 4. Conclusions

A “new” HT/O approach for the automated and parallel purification of polymers and polymer libraries via dialysis is proposed in this contribution. The developed APD setup consists of a customized dialysis rack to house several commercially available dialysis tubes, a dialysis container, a magnetic stirring system, and a pump, all installed within a commercially available APS. The setup allowed the efficient removal of residual monomers (and solvents) from homopolymer materials previously synthesized via RAFT polymerization. In addition, the developed APD method enabled the monitoring of the progress of the dialysis process of the different aqueous samples via automated sampling and followed by sample analyses with different techniques (i.e., ^1^H-NMR and SEC). The proposed APD approach can be rather easily adapted to automatically prepare samples for other HT/O analytical techniques, such as ultraviolet-visible, infrared, or RAMAN spectroscopies, dynamic light scattering, turbidity measurements in solution, etc. via the utilization of suitable sample vials, 96-well plates, etc., for screening the properties. Likewise, the APD method proposed herein can also be extended to the purification of copolymers and other chemical substances suitable for dialysis. It is worth mentioning that before introducing the described APD method, suitable chemical compatibility between the intended dialysis solvents, substances to be dialyzed, and commercially available dialysis membranes must be ensured by following up the recommendations of the corresponding membrane manufacturers. In this specific contribution, the described APD method can mostly be used for polymers that are soluble in methanol, ethanol, water, and/or mixtures derived thereof. However, other dialysis membranes with different chemical compatibility may be (or become) commercially available to apply the described APD method to other types of polymers or chemical substances. All in all, the APD method discussed in this contribution can be regarded as a good alternative for the “unattended” and reliable purification of polymers and polymer libraries (or other chemical substances) prepared in APS (Figure 5). This method may be useful for overcoming current limitations of HT/O synthesis of polymer libraries, where the purification of the generated materials currently represents a significant constraint for establishing fully automated experimental workflows necessary to advance towards the full digitalization of research and development of new polymers for diverse applications (Figure 5) [3]. The full automation of entire experimental workflows in combination with proper mathematical algorithms can enable the utilization of machine learning (ML) (and/or artificial intelligence (AI)) techniques for an accelerated and more efficient R&D process in polymer science as well as for the creation of autonomous or self-driving laboratories in the near future [47].

## Figures and Tables

**Figure 1 polymers-14-04835-f001:**
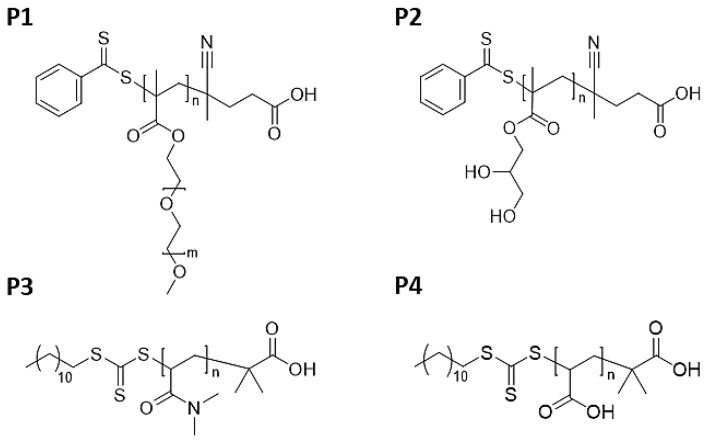
Schematic representation of the chemical structures of polymers P1 to P4 synthesized by RAFT polymerization.

**Figure 2 polymers-14-04835-f002:**
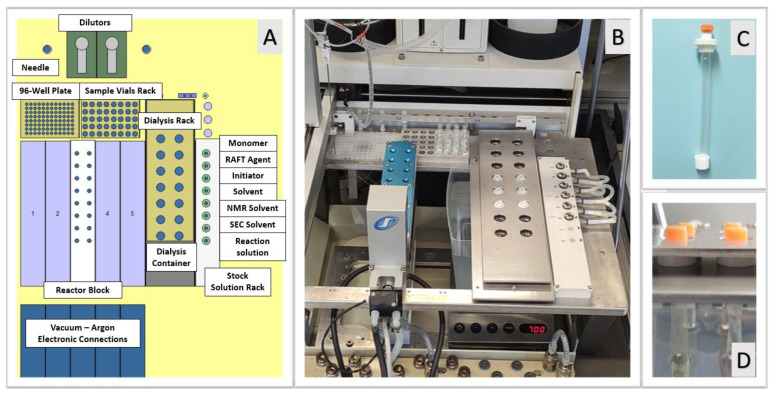
(**A**) Schematic representation of ASW2000 APS adapted with the APD system. Overviews of: (**B**) ASW2000 APS adapted with the APD system, (**C**) dialysis tubes used for the APD system, and (**D**) customized rack holding the dialysis tubes in the dialysis container.

**Figure 3 polymers-14-04835-f003:**
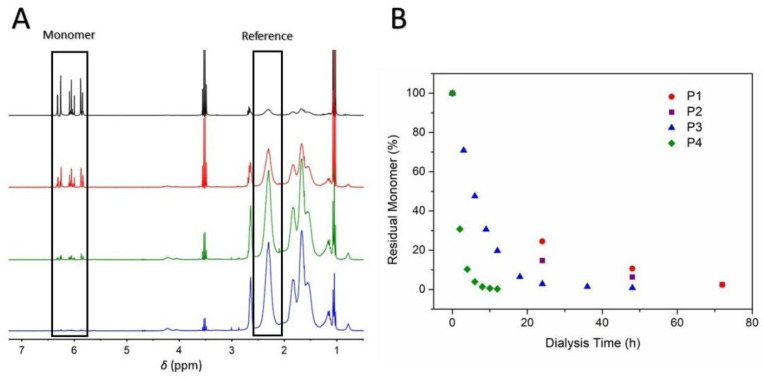
(**A**) ^1^H NMR spectra in D_2_O with a water suppression function for polymer P4 at different times to monitor the dialysis process. (**B**) Residual monomer vs. dialysis time for aqueous solutions of polymers P1–P4.

**Figure 4 polymers-14-04835-f004:**
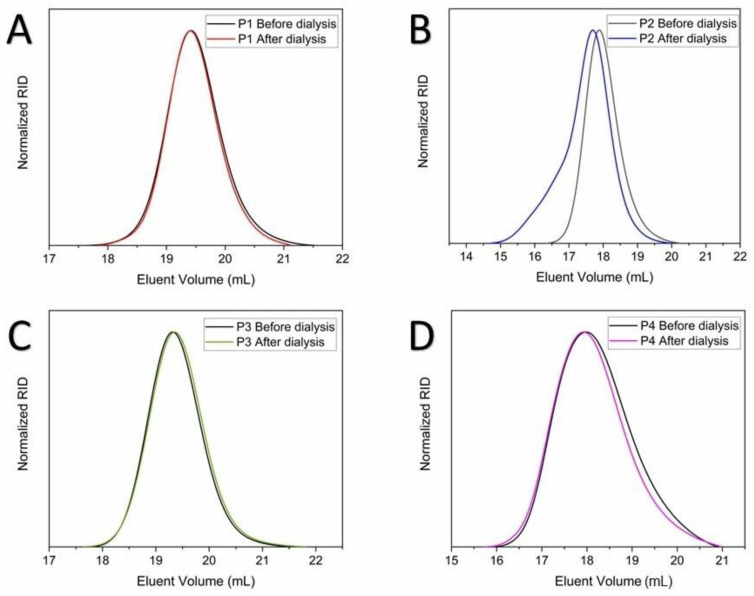
SEC traces of polymers P1-P4. (**A**–**C**) SEC eluent: DMAc + 0.21 wt. % LiCl, RI detection, calibrated against PMMA standards. (**D**) SEC aqueous eluent: 0.1 M NaNO_3_/0.05% NaN_3_, RI detection, calibrated against PEG standards.

**Figure 5 polymers-14-04835-f005:**
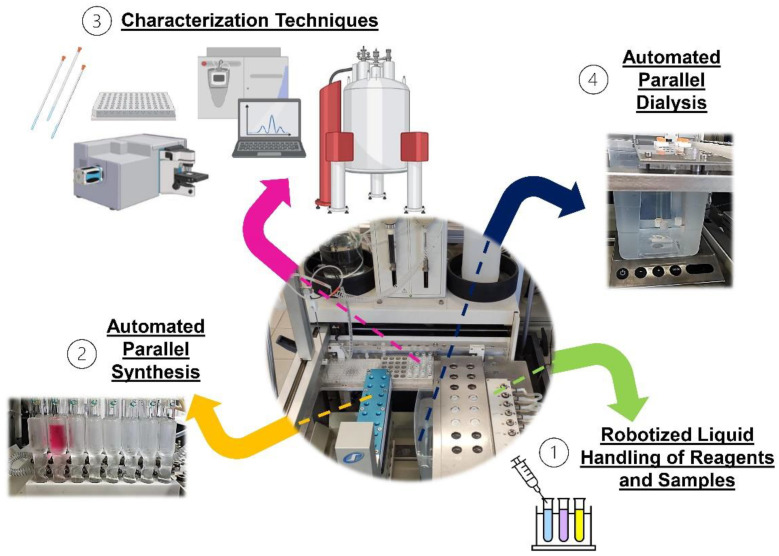
Schematic overview of the proposed APD method and its potential application in the synthesis and purification of polymer libraries utilizing commercially available APS.

**Table 1 polymers-14-04835-t001:** Summary of reaction conditions for the synthesis of polymers P1 to P4.

Entry	Monomer (M) /mmol	Chain Transfer Agent (CTA) /mmol	Initiator (I) /mmol	Solvent/mL	M:CTA:I Ratio ^a^	Temp. (°C)	Reaction Time (h)
P1	PEGMA /3.50	CPAD /0.1	AIBN /0.01	Ethanol /5.38	32:1:0.1	65	10
P2	GMMA /6.55	CPAD /0.06	ACVA /0.006	Ethanol /6.03	109:1:0.1	70	7
P3	DMA /13.94	DTMPA /0.08	AIBN /0.008	Dioxane /5.53	174:1:0.1	70	0.5
P4	AA /13.99	DTMPA /0.06	AIBN /0.006	Ethanol /6.04	233:1:0.1	65	3.5

(^a^) Note that different M:CTA:I ratios were selected for the different polymerization experiments in order to obtain polymers with a theoretical number average molar mass (M_n,theo_) close to 10 KDa at a monomer conversion of 60%.

**Table 2 polymers-14-04835-t002:** Dialysis conditions and physicochemical characterization of polymers P1–P4.

Entry	Conversion ^a^ (%)	Dialysis Time (h)	Residual Monomer ^a,b^ (%)	*M*_n_ theo. ^c^ (kg mol^−1^)	*M*_n_ ^d^ (kg mol^−1^)	*Ð* ^d^	*M*_n_ ^d^ (kg mol^−1^)	*Ð* ^d^
					Before Dialysis		After Dialysis	
P1	43	72	2.44	7.38	7.6	1.11	7.8	1.10
P2	55	93	1.33	9.34	19.1	1.11	25.4	1.26
P3	46	48	0.83	8.09	8.2	1.11	7.9	1.12
P4	51	12	0.32	8.79	11.5 ^e^	1.47 ^e^	12.3 ^e^	1.47 ^e^

(^a^) Monomer conversion and residual monomer were determined by ^1^H-NMR. (^b^) Calculated taken the residual monomer at start of dialysis as 100 % value. (^c^) Calculated from M_n,theo_ = ([[Monomer]_0_/[CTA]_0_ × Conv. M_M_] + M_CTA_), where [Monomer]_0_ and [CTA]_0_ are the initial concentrations of monomer and CTA, respectively. Conv. is the monomer conversion and M_M_ and M_CTA_ are the molar masses of monomer and CTA, agent respectively. (^d^) Determined by SEC, eluent DMAc + 0.21 wt. % LiCl, RI detection, calibrated against PMMA standards. (^e^) Determined by SEC, aqueous eluent 0.1 M NaNO_3_/0.05% NaN_3_, RI detection, calibrated against PEG standards.

## Data Availability

Not applicable.

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
