# Peer review of "Automated Parallel Dialysis for Purification of Polymers"

_polymers, 2022, doi:10.3390/polym14224835_

Round 1

Reviewer 1 Report

This study about "purification of polymers" gives good method. The automated parallel dialysis shows complete descriptions in system. Every instrument and method collocate each other. The authors show well-organized in this manuscript. 

This manuscript indeed creates detail and novel method for purification of polymers. The NMR spectrum also prove this method shows good.

But this manuscript still need to add a little information. 1. Although this method shows excellent, the authors need to list some previous and reported method to compare. Even the authors should list table to show advantages and disadvantages between this published method and previous reported method. The authors should write more these information in introduction

Due to this good method for  "purification of polymers" (NMR can prove), the authors should draw a diagram to explain the mechanism of this new method (this diagram may become graphical abstract).         

Author Response

Please see the attachment for a point-by-point response to the comments of both reviewers.

Reviewer 2 Report

Sanchez et al report on the automated synthesis and purification of four different polymers, synthesised by RAFT polymerisation and purified by dialysis. The work constitutes a focussed, well-presented study on chemical automation and should be of considerable interest to the polymer science community. The manuscript is well-written and the work is executed adequately. I recommend this manuscript for publication in Polymers, following consideration of the following minor points:

1.       Sugars and proteins are also polymers, so the sentence in the introduction concerning the use of dialysis for purifying polymers, as well as sugars and proteins, needs to be reworded slightly.

2.       The term ‘rapid’ for the purification is questionable. For the quickest polymer (PAA), it is 12 hours and the term rapid is misleading to the reader. Even though this is relatively fast for dialysis, a caveat of some kind is needed when using this term to describe the work (especially for the ones that took multiple days).

3.       Looking at Table 1, different degrees of polymerisation have been targeted (M:CTA ratios) for the four polymers. Is this to try to target polymers with the same final molar mass? In any case, a justification is needed for why these degrees of polymerisation were chosen.

4.       How much trioxane was added to the polymerisation solutions? Does it have any effect or influence on the rate of the polymerisation reaction?

5.       Some explanation as to why these CTAs were chosen for the different monomers would be useful to the reader.

6.       The authors mention the limitations of this approach being constrained to polymers that are soluble in methanol, ethanol, water and/or mixtures thereof, which is great and really important for this study. I would recommend including such a statement in the abstract and/or the conclusion section.

7.       The monomer conversions in the RAFT polymerisation reactions are quite low. Was this deliberate for this study? A comment to explain this would be useful. Moreover, would the dialysis (and subsequent analysis of residual monomer) approach be as effective for more typical RAFT systems where monomer conversion is often >80% or even >90%?

8.       There are a very small number of typographical/grammatical errors that need checking throughout the manuscript.

Author Response

(The authors gave the same response as above.)
